# Local and Systemic Antibody Responses in Beef Calves Vaccinated with a Modified-Live Virus Bovine Respiratory Syncytial Virus (BRSV) Vaccine at Birth following BRSV Infection

**DOI:** 10.3390/vetsci10010020

**Published:** 2022-12-29

**Authors:** David A. Martínez, Manuel F. Chamorro, Thomas Passler, Laura Huber, Paul H. Walz, Merrilee Thoresen, Gage Raithel, Scott Silvis, Ricardo Stockler, Amelia R. Woolums

**Affiliations:** 1Department of Clinical Sciences, College of Veterinary Medicine, Auburn University, Auburn, AL 36849, USA; 2Department of Pathobiology, College of Veterinary Medicine, Auburn University, Auburn, AL 36849, USA; 3Department of Pathobiology and Population Medicine, College of Veterinary Medicine, Mississippi State University, Starkville, MS 39762, USA; 4J.B. Taylor Veterinary Diagnostic Laboratory, Department of Agriculture and Industries, Alabama Veterinary Diagnostic Laboratory System, Elba, AL 36323, USA

**Keywords:** bovine respiratory syncytial virus, IgG-1, IgA, antibodies, vaccine

## Abstract

**Simple Summary:**

Bovine respiratory syncytial virus (BRSV) is a common cause of respiratory disease in calves. Vaccination of young calves against BRSV is a common prevention strategy; however, antibodies derived from maternal colostrum interfere with vaccine response and efficacy in young calves. The objective of this study was to determine if vaccination before colostrum absorption results in the effective induction of immune responses and clinical protection in calves. Within 6 h of birth, beef calves were assigned to 2 different treatment groups. Group Vacc (*n* = 25) was vaccinated with a modified-live virus (MLV) intranasal (IN) BRSV vaccine. Group Control (*n* = 25) remained unvaccinated. At approximately 3 months of age, calves were experimentally infected with BRSV. Immune responses and viral shedding were evaluated before and after infection. Respiratory signs before and after infection as well as viral shedding were similar between Vacc and Control calves. Local and systemic antibody responses were similar and suggested natural BRSV exposure before experimental infection. Based on the results from this study, early vaccination does not provide advantages for the clinical protection of calves from endemic BRSV farms.

**Abstract:**

Maternal antibodies interfere with BRSV vaccine responses and efficacy in young calves. The objective of this study was to determine if vaccination before the complete absorption of colostral antibodies results in adequate immune priming and clinical protection of beef calves. Within 6 h of life, calves were randomly assigned to 2 different treatment groups. Group Vacc (*n* = 25) received a single dose of a modified-live virus (MLV) BRSV vaccine intranasally (IN) and group Control (*n* = 25) received 2 mL of 0.9% saline IN. At approximately 3 months of age, all calves were experimentally challenged with BRSV. Serum and nasal secretion samples were collected before and after challenge for BRSV real-time RT-PCR and antibody testing. Respiratory signs were not observed before challenge. After challenge, respiratory scores were similar between groups. On the challenge day, >40% of calves in each group were febrile. The mean serum and nasal BRSV-specific antibody titers indicated natural BRSV exposure before the experimental challenge in both groups. All calves tested positive for BRSV and had a similar duration of shedding after challenge. Based on these results, vaccination at birth does not offer advantages for immune priming or clinical protection for beef calves in BRSV-endemic cow-calf herds.

## 1. Introduction

The bovine respiratory disease complex (BRDC) is the most common and economically important disease of beef calves in the United States (US). In young beef calves, the typical clinical presentation of BRDC is pre-weaning or nursing calf pneumonia. Losses associated with BRDC within this sector of the beef industry in the US have been estimated to be as high as $55 million per year [1,2]. Vaccination of calves against BRDC pathogens is a common strategy for the prevention of clinical disease; however, vaccination efficacy is variable in different production settings [3]. Bovine respiratory syncytial virus (BRSV) plays an important role in the pathogenesis of BRDC and pre-weaning calf pneumonia in beef herds [4]. Vaccination of young calves with intranasal (IN) modified-live virus (MLV) BRSV vaccines between 3 and 11 days of life has become a regular practice among producers and veterinarians to reduce clinical disease associated with BRSV infection in claves [5,6,7]; however, maternal antibodies present at the time of vaccination interfere with immune priming and compromise adequate antibody responses [8]. Transfer of colostral BRSV-specific immunoglobulin G-1 (IgG-1) into the upper respiratory tract of young calves not only could protect against infection but also block BRSV vaccine antigens from IN vaccination. Results from a previous study demonstrated considerable levels of nasal BRSV-IgG-1 at 48 h of life of beef calves that nursed colostrum from their dams [9]. Additionally, results from previous studies suggest that the duration of local antibody responses (i.e., BRSV-specific immunoglobulin-A) induced by vaccination of calves with IN MLV BRSV vaccines between 3 and 11 days of age is short-lived [7,10]. The interference by pronounced levels of colostral IgG-1 in the upper respiratory tract of young calves during the first week of life could prevent adequate immune priming and result in a short duration of local respiratory antibody responses, thus causing reduced efficacy of IN MLV BRSV vaccination at an early age. The first objective of this study was to determine if vaccination of beef calves with an IN MLV BRSV vaccine within 6 h of birth before complete absorption and transfer of colostral IgG-1 results in adequate priming and duration of nasal BRSV-IgA responses. The second objective was to determine if vaccination of beef calves at birth provides clinical advantages following experimental infection with BRSV. 

## 2. Materials and Methods

### 2.1. Experimental Design

Beef calves from a single herd born to dams vaccinated at least once with a multivalent inactivated-virus BRSV vaccine (Triangle 10^®^, Boehringer Ingelheim Animal Health USA Inc., Duluth, GA, USA) before calving were enrolled in this study. Using dam ID numbers, calves were randomly assigned to two different treatment groups before birth; group Vacc and group Control. After calving and within 6 h of birth, 50 bull calves from the Vacc and Control groups were randomly selected for the experimental portion of the study. Group Vacc (*n* = 25) received a single dose (2 mL) of an intranasal (IN) modified-live virus (MLV) BRSV vaccine (Inforce 3^®^, Zoetis Inc., Kalamazoo, MI, USA) following the manufacturer’s recommendations. Group Control (*n* = 25) received 2 mL of 0.9% phosphate-buffered saline (VeltivexTM^®^ sodium chloride injection solution 0.9%, Dechra Veterinary Products; Overland Park, KS, USA) intranasally. All calves nursed colostrum naturally from their dams, and cow-calf pairs from the Vacc and Control groups were placed in separate pastures with no fence-line contact for 60 days immediately after treatment administration. Following this period, pairs from both groups were placed together in the same pasture and creep feeders were introduced to allow calves to become familiarized with concentrate feed until early weaning. At approximately 3 months of age, all calves were weaned abruptly and transported 320 km to the experimental viral studies unit at Auburn University College of Veterinary Medicine for the challenge portion of the study. At arrival to the unit, calves were placed in a single pasture with ad libitum access to grass, fresh hay, water, and concentrate feed. Following an acclimation period of 5 days, all calves were challenged with BRSV by intranasal nebulization as described below. 

Colostrum samples from dams were collected at calving to evaluate immunoglobulin G (IgG) concentrations by colostrum Brix refractometry (MA871 Digital BRIX refractometer; Milwaukee instruments; Rocky Mount, NC, USA). At 48 h of age, all calves were weighed, identified (individual electronic tag [RFID] and regular ear tag), and castrated. Additionally, blood samples were collected from all calves to evaluate transfer of passive immunity by serum Brix refractometry. Additional blood and nasal secretion samples from calves were collected before and after BRSV challenge for antibody and virological assays. The Auburn University Institutional Animal Care approved the study and Use Committee (IACUC) PRN # 2019-3550 reviewed and approved all animal protocols. 

### 2.2. Experimental Challenge with BRSV

On Day 0, each calf was experimentally infected with 9 mL of a lung wash inoculum expanded once in Marvin Darby bovine kidney (MDBK) cells that contained 1 × 10^5^ TCID_50_ of BRSV strain GA-1/mL. Virus inoculation was performed by individual intranasal nebulization using an electronic nebulizer (Pulmo-Aide^®^ Compressor Nebulizer System, DeVilbiss Health Care; Washington, NY, USA) connected to a facemask (Era^®^ Equine Mask, BIOMEDTECH, Melbourne, VIC, Australia). 

### 2.3. Clinical Evaluation and Sample Collection

Clinical signs of disease (i.e., cough, nasal discharge, diarrhea, etc.) and treatments in study calves were recorded by on-farm personnel blinded to treatment allocation before arrival to the experimental viral studies unit and before challenge with BRSV. Following experimental challenge with BRSV, clinical evaluation and scoring was performed by a single veterinarian blinded to treatment allocation on days 0, 4, 6, 8, 10, 14, 21, and 28 relative to BRSV challenge. Blood samples were collected at 48 h and 30 days of life, and on days 0 and 28 after challenge for BRSV serum neutralization assays. Nasal secretion samples were collected at 48 h and 30 days of life, and on days 0, 4, 21, and 28 after challenge for BRSV IgG-1 and IgA determination. Additional nasal secretion samples were collected on days 0, 4, 6, 8, 10, 14, 21, and 28 after challenge to determine the presence of BRSV by real time reverse transcription polymerase chain reaction (RT-PCR). Samples from each calf were labeled to ensure that treatment allocation remained masked from personnel processing samples and performing the assays. 

On sampling days, clinical signs such as depression, rectal temperature, respiratory rate, cough, and nasal discharge were evaluated and a total respiratory score was assigned to each calf as previously described [11]. Clinical signs were scored on a scale from 0 to 3, where 0 was considered normal (no abnormalities noted) and 3 was the most abnormal finding. Depression was scored from 0 (bright alert responsive) to 3 (obtunded, recumbent, non-responsive), rectal temperature was scored from 0 (37.8–38.3 °C) to 3 (>39.4 °C), respiratory rate was scored from 0 (respiratory rate < 30 rpm) to 3 (respiratory rate > 100 rpm), cough was scored from 0 (none) to 3 (repeated spontaneous cough), and nasal discharge was scored from 0 (none or serous discharge) to 3 (purulent bilateral discharge). The sum of individual scores including rectal temperature, depression, respiratory rate, nasal discharge, and cough determined the presence of mild, moderate, or severe respiratory disease [11]. Briefly, mild respiratory disease was determined by a sum of individual scores between 0–5, moderate respiratory disease was determined by a sum of scores between 6–10, and severe respiratory disease when the sum of scores was >10. In addition to clinical evaluation, individual body weights were obtained at birth, on day 0 (BRSV challenge), and days 14 and 28 after BRSV challenge using a portable livestock electric scale (Livestock Platform Scale^®^ Brecknell, Fairmont, MN, USA) that was reseted to zero prior to and after each weighing. 

### 2.4. BRSV Neutralizing Antibodies in Serum

A virus neutralization assay for detection of BRSV neutralizing antibodies on sera was performed as previously described [12]. Samples were heat-inactivated in a water bath, then serial 2-fold dilutions starting from 1:10 to 1:1000 were performed in 96-microwell flat-bottom plates, followed by addition of 500 µL of 100 TCID_50_ BRSV to each well; all the corresponding samples dilution 3 microwells were inoculated with virus media. The plates were incubated at 37 °C in 5% CO_2_ for 1 h, then MBDK cell suspension with 7% bovine serum and an antibiotic/antimycotic solution containing streptomycin, penicillin, and amphotericin B were added. The plates were incubated under the same conditions for two weeks, and the cells were evaluated daily for evidence of cytopathic effect. Results from the test were reported as the inverse of the lowest dilution of serum required to inhibit all cytopathic effect; for the study analysis the results were Log2 transformed.

### 2.5. Determination of Anti-BRSV IgG-1 and IgA in Nasal Secretions

Bovine respiratory syncytial viral particles were inactivated with 2 µM binary ethyleneimine, neutralized with sodium thiosulfate, and diluted 1:800 in carbonate bicarbonate buffer (pH, 9.5). The resulting solution was used to coat microwells of 96-well polystyrene plates. After coating, the plates were incubated overnight at 4 °C and washed 3 times with PBS containing 0.05% polysorbate 20 (Tween^®^20, Sigma-Aldrich, St. Louis, MO, USA). After washing, 200 μL of PBSS solution containing 5% sheep serum albumin (Sheep serum albumin; Sigma-Aldrich; St. Louis, MO, USA) were added to each well for blocking, then the plates were incubated at 37 °C for 1 h followed by 3 washes. 

Vials containing nasal secretion samples were thawed and vortexed. Each sample was initially diluted 1 : 1 in Pluronic F127 and then diluted 1:100 in polysorbate 20. From this dilution, serial 2-fold dilutions were prepared up to 1:200 for IgG-1 and up to 1:1600 for IgA and each dilution was analyzed in triplicate (i.e., each sample dilution was added to 3 wells). If the coefficient of variation among the 3 values was >20%, the outlier value was removed, and the mean value of the 2 remaining samples was used in the calculation of the antibody titer. Samples with an optical density value that was too great to be measured accurately were tested again at a higher dilution. Samples with an optical density value that was too low to be measured accurately were tested again at a lower dilution with the lowest dilution tested being 1:25 for IgA and 1:100 for IgG-1. In addition to the samples, each plate had 3 microwells containing the following: positive control, which was a nasal secretion sample from a known BRSV antibody positive calf diluted 1:100 in polysorbate 20; negative control, which was low-IgG fetal bovine serum (FBS, Sigma-Aldrich; St. Louis, MO, USA) [diluted 1:100 in polysorbate 20]; and blank, which was polysorbate 20 alone. For BRSV-specific IgA, horseradish peroxidase-conjugated rabbit anti-bovine IgA (Rabbit anti-bovine IgA, Bio-Rad, Hercules, CA, USA) diluted 1:500 in an ELISA wash buffer (PBS + 0.05% TWEEN 20) and ABTS substrate solution (2,2′-azino-bis[3-ethylbenzothiazoline-6-sulphonic acid]) (ABTS, Sigma-Aldrich, St. Louis, MO, USA) were added to each well. For BRSV IgG-1, horseradish peroxidase-conjugated sheep anti-bovine IgG-1 (Sheep anti-bovine IgG-1 HRP; Bio-Rad, Hercules, CA, USA) diluted 1:7500 in an ELISA wash buffer (PBS + 0.05% TWEEN 20) and OPD substrate were added to each well. All plates were read by a plate reader set at a wavelength of 405 nm. Wells positive for anti-BRSV IgA and IgG-1 yielded a green (IgA) or yellow (IgG-1) product when the bound peroxidase-conjugated rabbit anti-bovine IgA and sheep anti-bovine IgG-1 reacted with the ABTS and OPD substrates, respectively. Immunoglobulin A and IgG-1 titers were reported as the inverse of last dilution that was ≥2 times the mean optical density value of the negative control. 

### 2.6. Real-Time Reverse Transcription PCR

Real-time reverse transcription polymerase chain reaction (RT-PCR) was performed on nasal secretion samples as previously described [13]. Briefly, nasal secretion sample aliquots were subjected to RNA extraction using RNAzol^®^ (Sigma-Aldrich, St. Louis, MO, USA) following manufacturers recommendations. Once extracted, the RNA templates were reverse transcribed and amplified with qScript™ XLT One-Step RT-qPCR ToughMix (Quantabio^®^, Beverly, MA, USA) using BRSV specific primers and probes [13]. Each reaction (2.5 μL) was performed in a BioRad CFX96^®^ (Bio-Rad^®^, Hercules, CA, USA) and results were analyzed by BioRad CFX manager^®^ (Bio-Rad^®^, Hercules, CA, USA). The detection limit of the assay was established at 10^1^ BRSV RNA copies/μL. 

### 2.7. Statistical Analysis

Data were analyzed using statistical software (RStudio (Version 1.4.1717); Boston, MA, USA). The normality of the data was assessed using the Shapiro–Wilk test and examination of the residuals. Data were analyzed using generalized mixed-effects models with animal ID as the random effect, immunoglobulin titers, virus neutralization titers, rectal temperatures, and body weights as dependent variables and vaccination status and experiment time as independent variables. Post hoc familywise comparisons were performed using Tukey–Kramer with Bonferroni correction. Kaplan–Meier curves were generated to display BRSV shedding via nasal secretions over time for Vacc versus Control calves. For non-parametric variables and proportions (i.e., Brix values, clinical scores, real time RT-PCR cycle threshold values), group comparisons were performed using the Fisher’s exact test, Chi-square (categorical predictor variables, 2 groups), Kruskal–Wallis (categorical predictor variables, >2 groups), and Wilcoxon Rank-Sum test (to compare medians between 2 independent populations). For all analyses, significance was set at *p*-value <0.05.

## 3. Results

### 3.1. Transfer of Passive Immunity and Clinical Outcomes

The median colostrum Brix value for dams of calves in the Vacc group (29.5%) was greater compared with the median Brix value for dams of calves in the Control group (26.7%); however, this difference was not statistically significant (*p* = 0.12). Similarly, the median serum Brix value for calves in the Control group (10.7%) was greater compared with that of calves in the Vacc group (10.4%); however, this difference was not statistically significant (*p* = 0.18). 

Clinical disease was not recorded during the pre-weaning period and before BRSV challenge. On the day of BRSV challenge, immediately prior to inoculation, 11 calves from the Control group and 9 calves from the Vacc group presented a fever (rectal temperatures ≥39.7 °C). Additionally, 2 calves from the Control group and 3 calves from the Vacc group coughed repeatedly and spontaneously. Following BRSV challenge, the mean total respiratory score increased in all calves from days 0 to 8; however, significant differences were only observed in Vacc calves at 6 and 8 days post-challenge. In contrast, the mean total respiratory score decreased in both Vacc and Control calves at 21 and 28 days post-challenge; however, statistically significant differences of mean respiratory scores between Vacc and Control calves were not detected at any time point during the study (Figure 1). The mean rectal temperatures in calves of both groups remained elevated during the first 14 days post-challenge; however, statistically significant differences between Vacc and Control calves were not detected at any time point during the study (Figure 2). On day 18 post-challenge, one Vacc group calf was found dead in the pasture with no previous clinical signs of disease. Necropsy and histopathology of this calf revealed severe fibrinonecrotizing cystitis, bilateral pyelonephritis, and diffuse bronchointerstitial pneumonia as possible causes of death. Lung tissue from this calf was determined to be negative for BRSV and other respiratory pathogens. 

There was a significant effect of time on the mean individual body weights (BW) of both, Vacc and Control calves. The mean individual BW significantly increased from birth to weaning in both groups; however, statistically significant differences between Vacc and Control calves were not detected at any time point during the study (Figure 3). The mean average daily gain (ADG) ± SEM from birth to Day 0 in Vacc and Control calves (1.2 ± 0.04 kg/d vs. 1.3 ± 0.03 kg/d, respectively) was not significantly different between groups (*p* = 0.48). The mean ADG ± SEM from Day 0 to Day 28 was greater in Vacc calves compared with Control calves (0.12 ± 0.05 kg/d vs. −0.05 ± 0.08 kg/d, respectively); however this difference was not statistically significant (*p* = 0.12). 

### 3.2. BRSV Neutralizing Antibodies in Serum

The mean ± SEM Log2 BRSV serum neutralizing antibody (SNA) titers from the baseline at 48 h of life to Day 28 (last study day) were not significantly different between Vacc and Control calves at any time point. A significant BRSV SNA titer decay was observed in all calves between 48 h and 1-month of age and in the Control group between Day 0 and Day 28 (Figure 4A). In contrast, a suspension of BRSV SNA decay was observed in Vacc and Control calves between 1-month of age and Day 0 (Figure 4A). 

### 3.3. BRSV-IgG-1 and IgA Titers in Nasal Secretions

The mean ± SEM nasal BRSV IgG-1 titers from the baseline at 48 h of life to Day 28 were not significantly different between Vacc and Control calves at any time point. The mean nasal BRSV IgG-1 titers in both groups were high at 48 h following colostrum intake; however, after 1-month of age nasal BRSV IgG-1 levels decayed to minimal levels for the reminder of the study (Figure 4B). In contrast, nasal BRSV IgA titers were undetectable in Vacc and Control calves at 48 h and 1-month of age, but significantly increased in both groups on Day 0 before experimental infection with BRSV (Figure 4C). Following Day 0, BRSV IgA titers significantly decayed in both groups until the end of study; however, statistically significant differences were not observed between Vacc and Control at any time point (Figure 4C). 

### 3.4. BRSV Real-Time RT-PCR in Nasal Secretions

Following BRSV challenge, nasal secretions samples from each calf in both treatment groups were confirmed to be positive to BRSV by real time RT-PCR at least once between days 4 to 10. The median and interquartile range of BRSV cycle threshold (CT) values detected by real time BRSV RT-PCR between calves from the Vacc and Control groups were not significantly different at any time point following BRSV challenge (Table 1). All calves were considered negative to BRSV by real time RT-PCR on days 0, 14, 21 and 28 post-challenge. The median number of days on which nasal secretions were positive for BRSV on real time RT-PCR was 2.17 days for Control calves and 2.36 days for Vacc calves, and this difference was not statistically significant (*p* = 0.84). Based on real time RT-PCR results in nasal secretion samples after challenge, the risk of BRSV shedding was not significantly different (*p* = 0.6) between Vacc and Control calves (Figure 5). Ten pools of nasal secretion samples from Day 0, each representing 5 calves and both treatment groups, were submitted to the Kansas State Veterinary Diagnostic Laboratory for multiplex PCR testing including several bovine respiratory pathogens (i.e., bovine viral diarrhea virus, bovine herpesvirus 1, BRSV, bovine coronavirus, influenza D virus, *Mannheimia haemolytica*, *Mycoplasma bovis*, *Pasterella multocida*, *Histophilus somni*, and *Bibersteinia trehalosi*). Two pools, one from the Vacc group and the other from the Control group, were test-positive for *Mannheimia haemolytica* and *Pasterella multocida* with no additional positive results to any other pathogen. 

## 4. Discussion

Specific BRSV immunoglobulin A (IgA) and immunoglobulin M (IgM) responses are detected in the respiratory tract of naïve young calves between 8 and 10 days following bovine respiratory syncytial virus (BRSV) infection or intranasal (IN) modified-live virus (MLV) vaccination [14,15]. In contrast, in calves with high levels of colostrum-derived serum antibody titers, BRSV IgA in the upper respiratory tract is undetectable in the period immediately following BRSV infection or vaccination [15]; however, despite the absence of an immediate upper respiratory BRSV IgA response following vaccination, anamnestic nasal IgA responses have been reported as early as 6 days following BRSV experimental infection [10,15]. In this study, nasal BRSV IgA was not detected following vaccination, but an anamnestic nasal BRSV IgA response was detected in calves from the Vacc and Control groups on Day 0 before experimental infection with BRSV. Additionally, a suspension of the expected decay of colostrum-derived BRSV serum-neutralizing antibody titers was observed between 1 month of age and challenge day. This suggests local and systemic activation of BRSV antibody production in study calves before experimental infection with BRSV. Results from previous studies demonstrated similar dynamics for bovine viral diarrhea virus (BVDV) serum-neutralizing antibody titers when a suspension in the decay of maternal immunity was observed in 2-month-old beef calves vaccinated with an MLV BVDV vaccine in the face of maternal antibodies (IFOMA) [16,17]. 

In this study, the dynamics of local and systemic BRSV antibody titers in study calves before experimental infection with BRSV suggest natural exposure to field or vaccine-origin BRSV antigens. It is possible that exposure to BRSV occurred after the commingling of calves in the same pasture 60 days after vaccination or sham vaccination and/or during transport prior to BRSV challenge. High seroprevalence rates of BRSV antibodies have been previously reported in beef cattle, and BRSV is considered endemic in some United States beef herds [4,18]. Results from one study suggested that changes in anti-BRSV serum antibody titers and seroconversion of unvaccinated cattle were the consequence of exposure to BRSV shed by subclinically infected cows or calves from the same herd [19]. It is possible that the stress of commingling or transport could have promoted BRSV shedding in subclinically infected animals in this study. Alternatively, transmission of vaccine-origin BRSV antigens from vaccinated calves to control calves following comingling could have resulted in acute infection and subsequent induction of immune responses; however, results from a previous study demonstrated nasal BRSV shedding from naïve calves vaccinated with the same vaccine used in the current study did not exceed 28 days [20]. Persistence and transmission of field and/or vaccine BRSV antigens among and between cattle from commercial beef herds using IN MLV BRSV vaccines is not well understood and should be a matter of further investigation. 

Clinical signs of respiratory disease (i.e., fever, cough) observed in Vacc and Control calves the day of challenge resembled a viral respiratory infection; however, attempts to identify BRSV and other viral respiratory pathogens in nasal secretion samples by real time RT-PCR on challenge day were unsuccessful. The presence of high levels of nasal BRSV IgA titers before experimental BRSV challenge strongly suggest a previous but recent natural BRSV infection before challenge. It is possible that by Day 0, the high titers of local antibody responses controlled a viral infection and eliminated viral particles from the upper respiratory tract preventing diagnosis. The presence of genetic material of *Mannheimia haemolytica* and *Pasterella multocida* in two pools of nasal secretion samples from Vacc and Control calves is difficult to interpret because these bacteria are considered normal commensals of the nasopharynx in healthy cattle [21]; however, respiratory signs observed in some study calves on the day of challenge could have been the result of bacterial pneumonia associated with virulent forms of these agents. Following BRSV challenge, clinical signs of respiratory disease were mild, and shedding as well as the viral load of BRSV in nasal secretions were not different between Vacc and Control calves. It is possible that the already elevated nasal BRSV IgA titers, present in all calves on the day of challenge, reduced viral replication, nasal shedding, and severity of clinical disease post-challenge. This is consistent with results from previous studies where a reduction in respiratory signs and nasal shedding was observed in calves with high levels of nasal BRSV IgA following experimental BRSV infection [10,15]. Additionally, other reports suggest that in cattle farms in which BRSV is endemic, repeated mucosal exposure to the virus over time increases the efficacy of local immune responses, preventing the virus from reaching the lungs and, therefore, severe respiratory disease [22,23]. The inadvertent natural exposure to BRSV before challenge prevented the evaluation of immune priming, duration of local antibody responses and clinical protection induced by vaccination at birth in this study; however, the absence of nasal BRSV IgA titers at 1-month of age in Vacc calves suggests that IN MLV BRSV vaccination as early as 6 h after birth was not sufficient to prevent maternal immunity interference. It is possible that high levels of BRSV IgG-1 transferred from maternal colostrum to the upper respiratory tract prevented priming and induction of detectable levels of BRSV IgA in Vacc calves following vaccination at birth. Results from a previous study demonstrated high levels of colostrum-derived nasal BRSV IgG-1 in calves that nursed colostrum from dams vaccinated with an inactivated-BRSV vaccine before calving [9]; however, nasal BRSV IgG-1 completely decayed before 1-month of age. This is consistent with the initial nasal BRSV IgG-1 titers and decay detected in Vacc and Control calves from this study and reflects the adequate transfer of passive immunity as demonstrated by serum Brix values [24,25]. Additionally, these results highlight the relatively short half-life of specific colostral antibodies (IgG-1) re-transferred to the upper respiratory mucosa in calves and their potential effect on IN MLV vaccination efficacy during this period.

## 5. Conclusions

Based on the results of this study, vaccination of beef calves with an IN MLV BRSV vaccine as early as 6 h after birth was not effective in overriding the interference from colostrum-derived antibodies and enhancing detectable BRSV-specific local antibody responses (i.e., nasal IgA) at 1-month of age. Additionally, natural exposure to BRSV may become equally effective as IN MLV vaccination at priming local and systemic immune responses in young calves from endemic beef cow-calf herds and preventing preweaning calf pneumonia as a consequence of BRSV infection. In cow-calf herds in which BRSV is not endemic vaccination of young calves may be necessary to prevent BRSV infection and disease; however, based on our results, we speculate that vaccination of young beef calves with MLV BRSV vaccines should be scheduled after 1 month of life to reduce maternal immunity interference and induce protective and possible long-lasting local and systemic immune responses.

## Figures and Tables

**Figure 1 vetsci-10-00020-f001:**
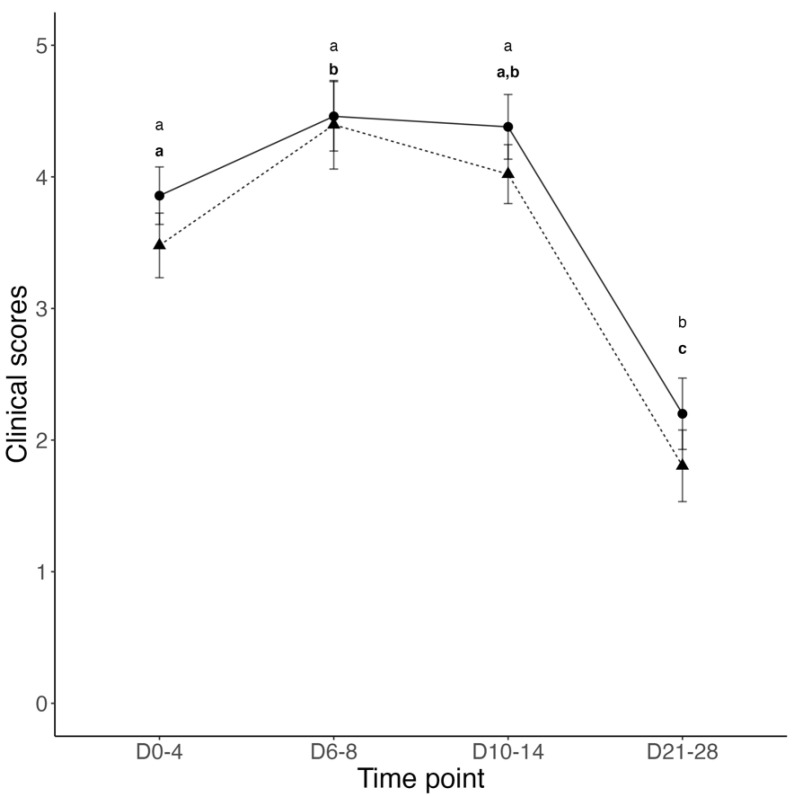
Mean ± SEM total respiratory scores on days 0 to 4, 6 to 8, 10 to 14, and 21 to 28 after challenge of calves vaccinated intranasally (IN) with a modified-live virus (MLV) BRSV vaccine in the first 6 h of life (dashed line and triangles) versus control calves (solid line and circles) subsequently challenged with BRSV. For each group and time point, the circle or triangle represents the mean, and the whiskers represent the SEM. A line for each group connects the group’s mean total respiratory score (on a scale of from 0 to 5 [none or mild respiratory disease] to >10 [severe respiratory disease]) progression throughout the study. Data were analyzed using generalized mixed-effects models. Distinct letters (small caps a, b, c) represent significant (*p* < 0.05) difference between time-points within participants of each group (Vacc in bold).

**Figure 2 vetsci-10-00020-f002:**
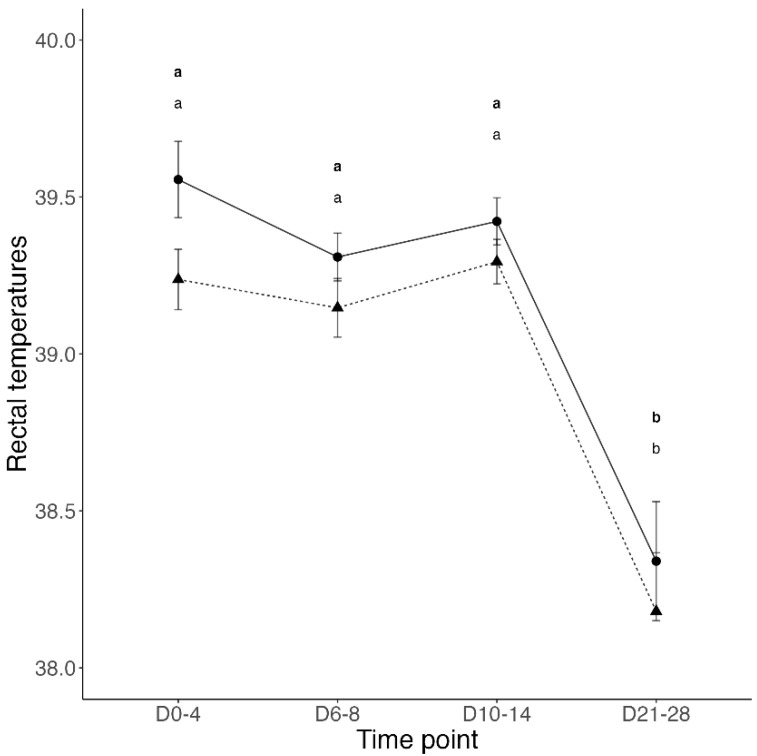
Mean ± SEM rectal temperature (°Celsius) during days 0 to 4, 6 to 8, 10 to 14, and 21 to 28 after challenge of calves vaccinated intranasally (IN) with a modified-live virus (MLV) BRSV vaccine in the first 6 h of life (dashed line and triangles) versus control calves (solid line and circles) subsequently challenged with BRSV. For each group and time point, the circle or triangle represents the mean, and the whiskers represent the SEM. A line for each group connects the group’s mean rectal temperature progression throughout the study. Data were analyzed using generalized mixed-effects models. Distinct letters (small caps a, b) represent significant (*p* < 0.05) difference between time-points within participants of each group (Vacc in bold).

**Figure 3 vetsci-10-00020-f003:**
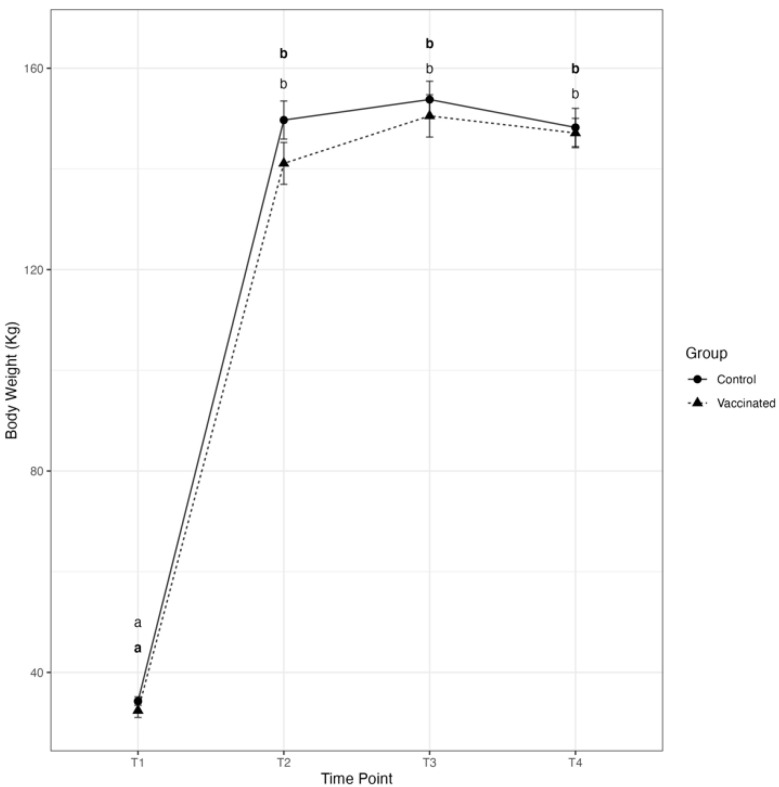
Mean ± SEM individual body weights at T1 (birth), T2 (Day 0), T3 (Day 14), and T4 (Day 28) of calves vaccinated intranasally (IN) with a modified-live virus (MLV) BRSV vaccine in the first 6 h of life (dashed line and triangles) versus control calves (solid line and circles) subsequently challenged with BRSV. For each group and time point, the circle or triangle represents the mean, and the whiskers represent the SEM. A line for each group connects the group’s mean body weight progression throughout the study. Data were analyzed using generalized mixed-effects models. Distinct letters (small caps a, b) represent significant (*p* < 0.05) difference between time-points within participants of each group (Vacc in bold).

**Figure 4 vetsci-10-00020-f004:**
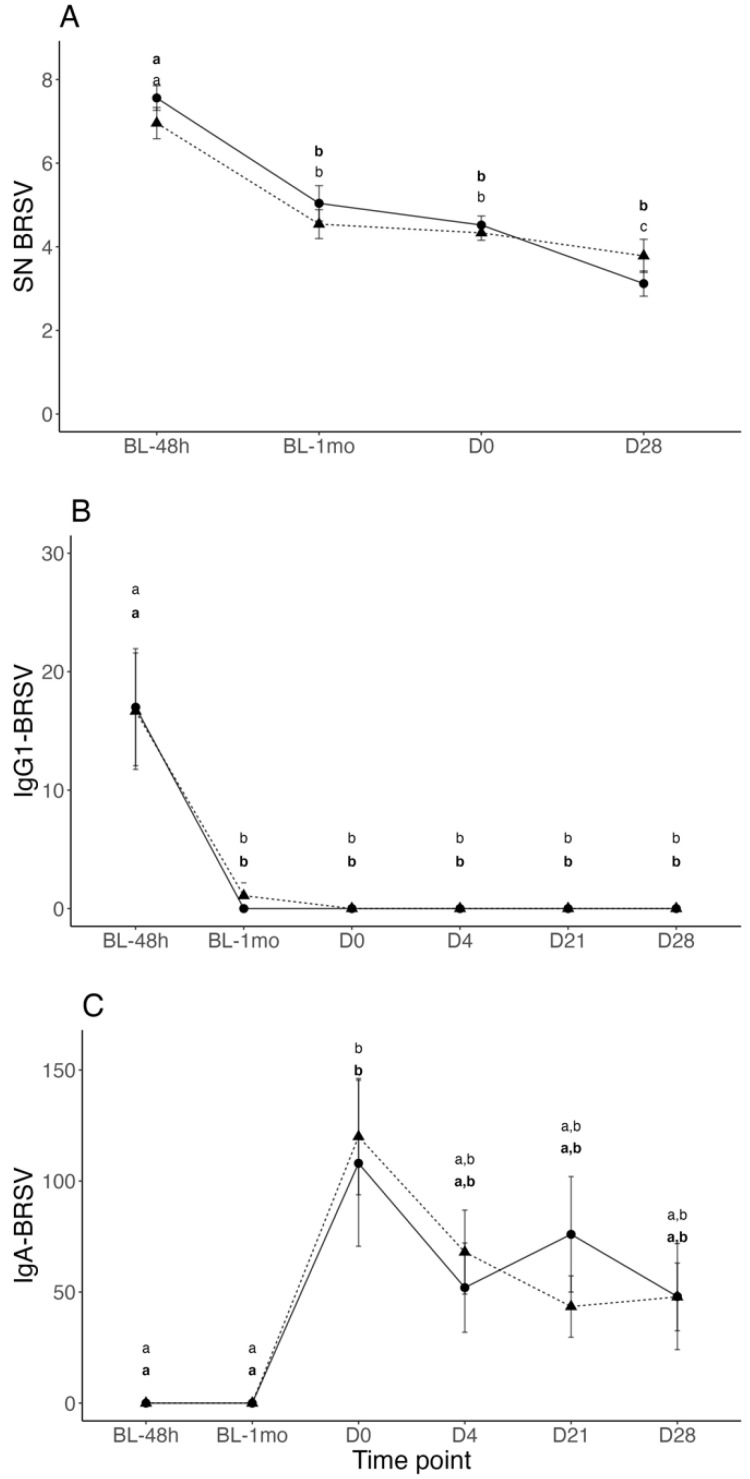
Mean ± SEM serum neutralizing ((**A**), log_2_ transformed), nasal BRSV immunoglobulin G1 (**B**), and nasal BRSV immunoglobulin A (**C**) antibody titers at 48 h of life (BL-48 h), 1-month of age (BL-1 mo), and days 0 [challenge day], 21, and 28 after BRSV challenge of calves vaccinated intranasally (IN) with a modified-live virus (MLV) BRSV vaccine in the first 6 h of life (dashed line and triangles) versus control calves (solid line and circles) subsequently challenged with BRSV. Data were analyzed using generalized mixed-effects models. Distinct letters (small caps a, b, c) represent significant (*p* < 0.05) difference between time-points within participants of each group (vaccinated in bold). Familywise multi comparisons were performed using Tukey–Kramer with Bonferroni correction. Statistically significant differences between groups were not observed at any time point.

**Figure 5 vetsci-10-00020-f005:**
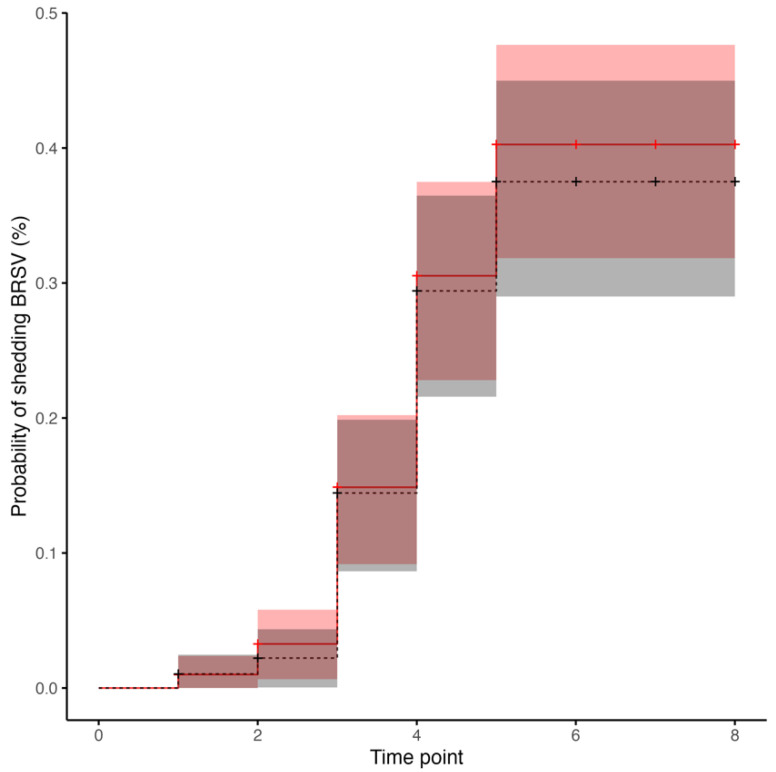
Kaplan–Meier curves of the cumulative probability of shedding Bovine Respiratory Syncytial virus (BRSV) detected by real time-RT-PCR assay on time intervals 0 to 8, in which each time interval is representing days 0 to 2, 2 to 4, 4 to 6, 6 to 8, 8 to 10, 10 to 14, 14 to 21, and 21 to 28 after BRSV challenge of calves vaccinated intranasally (IN) with a modified-live virus (MLV) BRSV vaccine in the first 6 h of life (dashed line and triangles) versus control calves (solid line and circles) subsequently challenged with BRSV. Tick marks represent the end of each time period, each step represents detection events of BRSV shedding, and shading represents the respective 95% CI for the probability of shedding BRSV by vaccinated (red) versus control (gray) calves.

**Table 1 vetsci-10-00020-t001:** Median and interquartile range (IQR) of cycle threshold (CT) values for BRSV in nasal secretion samples detected by real time-RT-PCR assay for Vacc versus Control calves as described in Figure 1 from challenge day (Day 0) to Day 28 of the study.

Time Point	BRSV Real Time RT-PCR CT (Median, (IQR))		*p*-Value
	Control	Vacc	
Day 0 (Challenge Day)	0	0	1.0
Day 4	32.7 (29.1, 34)	32.7 (31, 34.3)	0.85
Day 6	33.7 (32.2, 34.9)	34.5 (32.3, 36.2)	0.13
Day 8	32.9 (31.1, 34.1)	33.9 (32.1, 35)	0.36
Day 10	36.1 (33.2, 36.9)	34.9 (33.8, 36.8)	0.09
Day 14	0	0	1.0
Day 21	0	0	1.0
Day 28	0	0	1.0

## Data Availability

The data presented in this study are available on request from the corresponding author.

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
