# Peer review of "Local and Systemic Antibody Responses in Beef Calves Vaccinated with a Modified-Live Virus Bovine Respiratory Syncytial Virus (BRSV) Vaccine at Birth following BRSV Infection"

_vetsci, 2022, doi:10.3390/vetsci10010020_

Round 1
Reviewer 1 Report
The study by Martinez et al. targets the question, if the frequently practiced vaccination of beef calves soon after birth with a MLV BRSV vaccine affords increased protection against BRSV infection after weaning or is otherwise advantageous. This objective is clearly stated. The study is well designed and pays attention to many details that might influence the outcome (e.g. colostrum quality, uptake of colostrum, ….). Blinded evaluation increases the objectivity. The presentation of results is supported by diagrams.
Concerning these diagrams, I would prefer box plots over the line diagrams chosen, because lines suggest trends between data points which is not supported by measurements.
The authors conclude that there is no advantage in the early vaccination; however, they have to make the reservation that is applies to an endemic BRSV situation and may otherwise be different.
Author Response
Comments to the Author
The study by Martinez et al. targets the question, if the frequently practiced vaccination of beef calves soon after birth with a MLV BRSV vaccine affords increased protection against BRSV infection after weaning or is otherwise advantageous. This objective is clearly stated. The study is well designed and pays attention to many details that might influence the outcome (e.g. colostrum quality, uptake of colostrum, ….). Blinded evaluation increases the objectivity. The presentation of results is supported by diagrams.
Concerning these diagrams, I would prefer box plots over the line diagrams chosen, because lines suggest trends between data points which is not supported by measurements.
We appreciate the reviewer’s recommendation. We agree with the reviewer in that lines suggest data trends; however, we respectfully disagree with the reviewer in that trends are not supported by our measurements. We strongly believe that in this case our statistical analysis predicted the effects of time-points and of group (Vacc versus Control) on the outcomes. For most graphs, we did not find an effect of group, but we did find effects of time. Therefore, we believe line graphs would be the best way to represent our results and would like to keep them in our manuscript.
The authors conclude that there is no advantage in the early vaccination; however, they have to make the reservation that is applies to an endemic BRSV situation and may otherwise be different.
We thank the reviewer for the insightful comment and suggestion. We agree with the reviewer in that our conclusion on vaccination of calves at birth may only apply to individuals of this herd and to herds where BRSV is endemic but not to herds where BRSV is not endemic. We have modified the Conclusion section of the text between lines 516 and 521 of the revised manuscript to reflect the reviewer’s comment and suggestion.

Reviewer 2 Report
Comments for the authors
The manuscript entitled “Local and systemic antibody responses in beef calves vaccinated with a modified-live virus bovine respiratory syncytial virus (BRSV) vaccine at birth following BRSV infection” showed that vaccination of beef calves with an IN MLV BRSVvaccine as early as 6 hours after birth was not effective in overriding the interference from colostrum-derived antibodies and enhancing detectable BRSV specific local antibody responses at 1-month of age under normal farm conditions. The results provided useful information to prevent BRSV. However, there are some concerns in the manuscript as follows:
1. Please explain why choose the 3-month-old calf for the BRSV challenge in this study.
2. Before the challenge, many beef in both the Vaccine and Control groups had shown respiratory symptoms, why did they were still challenged with BRSV?
3. Serotypes 1 and 6 Mannheimia haemolytica, serotypes A and B Pasteurella multocida are pathogenic to cattle, and Mannheimia haemolytica and Pasteurella multocida were detected from two pools, respectively. Please analyze whether the positive samples is related to the sick cattle before challenge.
4. Please change the RT-PCR in the manuscript to real time RT-PCR, and report the detection limit of this assay.
5. Because the authors used real-time RT-PCR assay to detect BRSV, I suggest adding a table of CT values for post-challenge calves and comparing viral load in different groups.
6. Lines 367-370: check this sentence.
Author Response
The manuscript entitled “Local and systemic antibody responses in beef calves vaccinated with a modified-live virus bovine respiratory syncytial virus (BRSV) vaccine at birth following BRSV infection” showed that vaccination of beef calves with an IN MLV BRSVvaccine as early as 6 hours after birth was not effective in overriding the interference from colostrum-derived antibodies and enhancing detectable BRSV specific local antibody responses at 1-month of age under normal farm conditions. The results provided useful information to prevent BRSV. However, there are some concerns in the manuscript as follows:
- Please explain why choose the 3-month-old calf for the BRSV challenge in this study.
We thank the reviewer for the insightful question on the time of BRSV challenge with respect to calf vaccination in this study. We performed BRSV challenge in study calves at 3 months of age to replicate a typical case of pre-weaning or “summer” beef-calf pneumonia in cow-calf operations in the United States (US), which usually affects nursing calves between 2 and 4 months of age. Our hypothesis was that vaccination of calves with an intranasal (IN) modified-live virus (MLV) BRSV vaccine shortly after birth and before complete absorption and transfer of colostral immunoglobulins resulted in better priming and duration of memory local and systemic antibody responses to BRSV and therefore could provide clinical protection later in life.
- Before the challenge, many beef in both the Vaccine and Control groups had shown respiratory symptoms, why did they were still challenged with BRSV?
We thank the reviewer for this important question. The presence of respiratory signs in some calves before experimental BRSV challenge denoted the potential of clinical bovine respiratory disease (BRD) in those calves and BRD before BRSV challenge negatively impacted the hypothesis, objectives and results of our study. The decision to continue with BRSV challenge in all calves including those presenting signs of respiratory disease on challenge day was to replicate what would occur in a natural outbreak of BRD in nursing beef calves in which more than one infectious agent is usually involved and calves can be exposed and re-exposed to infectious agents at different time points. Additionally, since calves presenting respiratory symptoms on challenge day were from both, the Vacc and Control groups, we wanted to compare the effect of vaccination at birth on its ability to provide clinical advantages on calves potentially exposed to multiple BRD agents.
- Serotypes 1 and 6 Mannheimia haemolytica, serotypes A and B Pasteurella multocidaare pathogenic to cattle, and Mannheimia haemolytica and Pasteurella multocida were detected from two pools, respectively. Please analyze whether the positive samples is related to the sick cattle before challenge.
We thank the reviewer for this insightful comment. We agree with the reviewer in that the presence of two nasal-secretion pool samples positive to Mannheimia haemolytica and Pasteurella multocida could have indicated the presence of bacterial pneumonia on challenge day in some study calves, and this could have explained the high fevers, cough, and nasal discharges observed that day; however, in our opinion it is difficult to make the case of bacterial pneumonia without confirmation of the presence of bacteria in the lungs with associated lung pathology. Additionally, genetic characterization/typification of the Mannheimia haemolytica and Pasteurella multocida positive pool-samples was not performed and these bacteria are considered normal commensals of the upper respiratory tract of cattle. We have modified the text between lines 478 and 483 of the revied manuscript to reflect the reviewer’s concern.
- Please change the RT-PCR in the manuscript to real time RT-PCR, and report the detection limit of this assay.
We thank the reviewer for the correction. We have modified the text on lines 34, 127, 199, 200, 378, 380, 382, 384, 386, 387, 402, 426, and 473 of the revised manuscript to reflect the reviewer’s recommendation. Additionally, we have added to the text of the Materials and Methods section the detection limit of the BRSV real time RT-PCR assay between lines 205 and 208 of the revised manuscript to reflect the reviewer’s request.
- Because the authors used real-time RT-PCR assay to detect BRSV, I suggest adding a table of CT values for post-challenge calves and comparing viral load in different groups.
We thank the reviewer for the suggestion. We have added a table with the median and interquartile range of BRSV real time RT-PCR cycle threshold values of study calves post-challenge (Table 1) between lines 399 and 403 of the revised manuscript. Additionally, we compared the viral load among calves from the Vacc and Control groups; however, statistically significant differences among treatment groups were not observed at any time point. We have modified the text between lines 381 and 383 and 484 and 485 of the revised manuscript to reflect the reviewer’s recommendation.
- Lines 367-370: check this sentence.
We thank the reviewer for the recommendation. We have shortened the sentence to reflect the reviewer’s recommendation. Changes of the text can be found between lines 368 and 373 of the revised manuscript.

Reviewer 3 Report
The study is interesting, in my opinion the only uncontrolled aspect concerns the possible natural exposure to the BRSV virus which is supposed to have occurred before the challenge (experimental infection). This aspect makes the experimentation protocol slightly weaker but in any case this aspect is repeatedly reported by you and this allows the reader to make a correct and unbiased analysis. Overall, the results are interesting and worthy of being published.
Author Response
The study is interesting, in my opinion the only uncontrolled aspect concerns the possible natural exposure to the BRSV virus which is supposed to have occurred before the challenge (experimental infection). This aspect makes the experimentation protocol slightly weaker but in any case this aspect is repeatedly reported by you and this allows the reader to make a correct and unbiased analysis. Overall, the results are interesting and worthy of being published.
We thank the reviewer for the comment. We agree with the reviewer in that potential natural exposure to BRSV in our study negatively affected our hypothesis, objectives, and experimental design; however, this was something unexpected.

Round 2
Reviewer 2 Report
-